# Tick species diversity and potential distribution alternation of dominant ticks under different climate scenarios in Xinjiang, China

**Rui Ma**[1], **Chunfu Li**[1], **Ai Gao**[1], **Na Jiang**[1], **Jian Li**[1,2]\*, **Wei Hu**[1,3]\*, **Xinyu Feng**[4,5]\*

**1** College of Life Sciences, Inner Mongolia University, Hohhot, China, **2** Basic Medical College, Guangxi University of Chinese Medicine, Nanning, Guangxi, China, **3** Department of Infectious Diseases, Huashan Hospital, State Key Laboratory of Genetic Engineering, Ministry of Education Key Laboratory for Biodiversity Science and Ecological Engineering, Ministry of Education Key Laboratory of Contemporary Anthropology, School of Life Sciences, Fudan University, Shanghai, China, **4** School of Global Health, Chinese Center for Tropical Diseases Research, Shanghai Jiao Tong University School of Medicine, Shanghai, China, **5** One Health Center, Shanghai Jiao Tong University-The University of Edinburgh, Shanghai, China

\* leejianshin@163.com (JL); huw@imu.edu.cn (WH); fengxinyu2013@163.com (XF)

**Data Availability Statement:** All relevant data are within the manuscript and its Supporting information files.

## Abstract

Ticks are a hematophagous parasite and a vector of pathogens for numerous human and animal diseases of significant importance. The expansion of tick distribution and the increased risk of tick-borne diseases due to global climate change necessitates further study of the spatial distribution trend of ticks and their potential influencing factors. This study constructed a dataset of tick species distribution in Xinjiang for 60 years based on literature database retrieval and historical data collection (January 1963-January 2023). The distribution data were extracted, corrected, and deduplicated. The dominant tick species were selected for analysis using the MaxEnt model to assess their potential distribution in different periods under the current and BCC-CSM2.MR mode scenarios. The results indicated that there are eight genera and 48 species of ticks in 108 cities and counties of Xinjiang, with *Hyalomma asiaticum*, *Rhipicephalus turanicus*, *Dermacentor marginatus*, and *Haemaphysalis punctatus* being the top four dominant species. The MaxEnt model analysis revealed that the suitability areas of the four dominant ticks were mainly distributed in the north of Xinjiang, in areas such as Altay and Tacheng Prefecture. Over the next four periods, the medium and high suitable areas within the potential distribution range of the four tick species will expand towards the northwest. Additionally, new suitability areas will emerge in Altay, Changji Hui Autonomous Prefecture, and other local areas. The 60-year tick dataset in this study provides a map of preliminary tick distribution in Xinjiang, with a diverse array of tick species and distribution patterns throughout the area. In addition, the MaxEnt model revealed the spatial change characteristics and future distribution trend of ticks in Xinjiang, which can provide an instrumental data reference for tick monitoring and tick-borne disease risk prediction not only in the region but also in other countries participating in the Belt and Road Initiative.

**Funding:** This work was supported by the Inner Mongolia Autonomous Region Science and Technology leading talent team: Zoonotic disease prevention and Control Technology Innovation team (2022SLJRC0023 to WH); Key Technology Project of Inner Mongolia Science and Technology Department (2021GG0171 to WH); State Key Laboratory of Reproductive Regulation and Breeding of Grassland Livestock (2020ZD0008 to WH); Study on pathogen spectrum, temporal and spatial distribution and transmission features of the important emerging and re-emerging zoonosis in Inner Mongolia Autonomous Region (U22A20526 to WH) National Parasitic Resources Center (NPRC-2019-194-30 to WH). The funders had no role in study design, data collection and analysis, decision to publish, or preparation of the manuscript.

**Competing interests:** The authors have declared that no competing interests exist.

## Author summary

Tick-borne diseases pose a significant threat to public health worldwide, and their spread is being intensified by global climate change. The need for further research into the spatial distribution and influencing factors of ticks is becoming increasingly urgent. Our study compiled a comprehensive 60-year dataset on tick species distribution in Xinjiang to analyze trends and predict future distributions.

The study identified eight genera and 48 tick species across 108 cities and counties in Xinjiang, with *Hyalomma asiaticum*, *Rhipicephalus turanicus*, *Dermacentor marginatus*, and *Haemaphysalis punctatus* being the four dominant species. In addition, our study forecasts a northward expansion of suitable habitats for these species towards Altay and Tacheng Prefecture, with new suitability areas expected in Altay and the Changji Hui Autonomous Prefecture. The findings of this study provide a crucial map of tick distribution and offer valuable insights into future trends, serving as a valuable resource for tick monitoring and disease risk prediction in Xinjiang, particularly in light of the devastating effects of tick-borne diseases on human health.

## Introduction

The emergence of infectious diseases such as Ebola, SARS, MERS, and COVID-19 in recent years has brought the concept of global health to new heights, highlighting the interconnectedness of health challenges that are not confined within national boundaries [1]. As such, collaborative efforts on a worldwide scale have become increasingly necessary. China, in particular, has emerged as a pivotal player in the global health arena, actively addressing these challenges and contributing significantly to international health initiatives [2]. To foster economic cooperation and mutual development among participating countries, China proposed the Belt and Road Initiative (BRI) [3]. At the heart of the Silk Road Economic Belt, Xinjiang plays a vital role in facilitating trade between China, Central Asian countries, and Europe [4]. Consequently, the region has experienced increased international personnel exchanges and livestock trade. However, comprehensive studies on tick distribution and tick-borne diseases in this region remain limited. Therefore, further research is needed to fully understand the potential health risks associated with this increased trade and exchange.

Xinjiang's climate is characterized by significant diurnal temperature variations, low precipitation, and arid conditions typical of a temperate continental climate [5]. The region boasts diverse ecosystems, including forests, shrublands, grasslands, and semi-desert zones, providing favorable conditions for tick proliferation [6]. Currently, ticks in Xinjiang represent approximately one-third of the species identified in China, including unique species such as *Anomalohimalaya lotozkyi* and *Dermacentor pavlovskyi* [7]. Xinjiang is a hotspot for tick-borne diseases, with reported cases of Crimean-Congo hemorrhagic fever, Q-fever, and Lyme disease, among others [8]. Notably, neighboring countries along the BRI have also reported outbreaks of tick-borne diseases. For instance, there have been records of outbreaks and epidemics of tick-borne encephalitis, tularemia, tick-borne rickettsiosis, and Lyme disease in Kazakhstan and other neighboring countries [9]. Special attention should be paid to the recent reports of new tick-borne pathogens, such as the novel tick-borne virus, TBEV-2871 isolated from *Ixodes pavlovskyi* in Sergey E Tkachev et al. in Western Siberia, Russia, and Tyulek virus isolated by L'Vov DK et al. in Kyrgyzstan [10, 11]. Thus, it is of great significance to investigate the distribution patterns of ticks and the potential for tick-borne diseases to spread to

countries along the One Belt One Road initiative. This knowledge is crucial for ensuring the stability of these nations' economies and politics, as well as for the advancement of global health.

Previous studies show that the distribution of ticks is closely related to the natural environment and exhibits distinct regional and seasonal characteristics. Climate factors such as temperature, humidity, and precipitation can influence the growth and development of ticks, range of activity, vector capacity, and the transmission of associated pathogens they carry [12–17]. Therefore, assessing how climate, environment, hosts, and other factors impact the geographic distribution and population dynamics of ticks is one of the important directions in tick and tick-borne disease research. Currently, the existing literature on tick species research in Xinjiang mainly focuses on local investigations of certain specific tick species' types and distribution. Understanding the spatial characteristics of tick distribution in the entire region and its influencing factors is still very limited. Consequently, the primary objectives of this study are to identify possible distributions of tick species in Xinjiang over a period of 60 years and to construct potential spatial distribution models for the current and future dominant tick species in Xinjiang. Additionally, the study aims to analyze the key environmental factors that influence the spatial distribution of ticks in Xinjiang and predict future changes in suitable habitats. The research findings have significant implications for predicting the population distribution of ticks in Xinjiang, as well as for cross-border transmission, monitoring, and risk assessment of tick-borne diseases in both local areas and countries along the Belt and Road Initiative.

## Materials & methods

### Tick geographic distribution data collection and sorting

To retrieve literature on the geographic distribution of tick species in Xinjiang, China, we used the keywords "Xinjiang" and "tick" to search the CNKI (China National Knowledge Infrastructure) databases (the largest and most comprehensive academic database for Chinese scholarly publications.), Wanfang Data, Baidu Scholar, PubMed, and Google Scholar. The search query covered a time span from January 1963 to January 2023. By reading the titles, abstracts, and full texts of the articles, we selected those that included information on the geographic distribution of ticks in Xinjiang and provided their geographical coordinates or location information. If the latitude and longitude were not specified in the text, the coordinates of the distribution points were determined using the coordinate picking function in Google Maps based on the geographical location indicated in the text. To avoid overfitting caused by excessive concentration of distribution points, ArcGIS 10.4 software was used to set a buffer zone with a radius of 10 km for each obtained distribution point (consistent with the resolution of environmental climate data at 10 km). Only one distribution point was retained in each buffer zone.

### Source and selection of data on environmental variable

This study considered 25 environmental variables, including bioclimatic factors, topography, geographical landscape, and socio-cultural factors. The bioclimatic variables were derived from the Worldclim database, which provides data on 19 bioclimatic variables. The resolution of the 19 bioclimatic layers is 5 arc·minute-1 (approximately equivalent to a pixel grid of 10 km × 10 km near the equator). To avoid overfitting the climate model due to multicollinearity among climate variables, a resampling technique was applied to the 19 climate factors (bio1 to bio19) using the sampling function in ArcGIS 10.4 software. The resampled data was then imported into SPSS software for correlation analysis using Pearson's matrix. A correlation coefficient ($|r|$) greater than 0.9 was defined as high correlation, and the variables with high

correlation were filtered based on their contribution to the environmental factors. For a more compatible future climate change trend in China, the CMIP6 (Climate Model Intercomparison Project Phase 6) high-resolution climate BCC-CSM2.MR scenario under SSP245 (Shared Socioeconomic Pathways) scenario was projected for four future climate data periods: 2021–2040, 2041–2060, 2061–2080, and 2081–2100.

While climate plays a crucial role in the distribution of ticks, alterations in terrain characteristics, vegetation types, and host activity range within the distribution area can also have a significant impact on tick abundance and the transmission of vector-borne pathogens. Therefore, we selected major terrain variables, including slope, aspect, and elevation. We downloaded elevation data from the Geographic Spatial Data Cloud (www.gscloud.cn) website and used ArcGIS 10.4 software to calculate slope and aspect variables. The study simultaneously selected geographical landscape factors, including normalized difference vegetation index (NDVI) and land cover types (LC) obtained from the Resource and Environment Data Cloud Platform of the Chinese Academy of Sciences. This dataset includes several major land use types: cropland, forest, grassland, shrubland, urban fabric, and water surfaces. The normalized difference vegetation index (NDVI) data was originated from the Geospatial Data Cloud Platform (http://www.gscloud.cn). In addition, we have considered population density (PD) as an anthroposociological factor in our research. The data for population density was accessed from WorldPop (https://www.worldpop.org/). The basic map data used in the study was obtained from the Department of Natural Resources Standard Map Service System (https://www.webmap.cn/).

## Models constructed using the maximum entropy algorithm

The Maximum Entropy (MaxEnt) model is a widely used methodology in ecological niche modeling that relies on the fundamental theory of species ecological niches. The model utilizes information on the spatial distribution of species occurrence data and relevant environmental background data to estimate the probability distribution with the maximal entropy value, thereby providing an estimation of the potential distribution of a given species. The selected tick distribution points were imported into the MaxEnt model along with environmental variables. A random selection of 75% of the tick distribution points was designated as the training set, while the remaining 25% of points were assigned as the testing set to validate the model's accuracy. The following parameters were set: the available features were set to automatic features, including linear, quadratic, product, threshold, and hinge; the convergence threshold of the system was limited to $10^{-5}$, the maximum number of iterations was set to 500, and the output was configured as logical output. This allowed the model to generate a continuous mapping where the predicted distribution probabilities ranged between 0 and 1. In addition, the Jackknife procedure was conducted to evaluate the weights of various ecological factors. A univariate response curve was established to determine the suitable range of values for environmental variables based on their distribution probabilities. Based on the parameter settings, a series of model calculations were performed on four dominant tick species in Xinjiang using a bootstrap approach in MaxEnt. This process was repeated 10 times to ensure accuracy and reliability. The resulting data was then imported into ArcGIS 10.4 software, and the suitability areas for the four dominant tick species were classified using the "natural breaks (Jenks)" method.

Upon establishing the model, its predictive accuracy was assessed based on the area under the receiver operating characteristic curve (ROC curve). The evaluation criteria for the model were determined by the AUC values, which were classified as follows: AUC values ranging from 0.5 to 0.6 were deemed as failure, values from 0.6 to 0.7 as poor, values from 0.7 to 0.8 as

fair, values from 0.8 to 0.9 as good, and values from 0.9 to 1.0 as excellent. Notably, the accuracy of the model's predictions and the degree of correlation between environmental factors and species distribution was directly proportional to the AUC value. The closer the AUC value is to 1, the more accurate the model's predictions will be, which indicates a higher correlation between environmental factors and species distribution.

## Spatial variation of the centroid

We analyzed the spatial variation of the centroids of overall suitable habitats of the dominant tick species under different climate change scenarios, including the near current and four future periods. ArcGIS 10.4 software was used to convert the distribution map of suitability areas of the MaxEnt model to obtain the centroid coordinates after binary conversion. The spatial change direction and distance of centroid in suitability areas of dominant ticks were obtained by connecting centroid points under different climate conditions.

## Results

### Distribution of tick species in Xinjiang region

The present study conducted a comprehensive literature search and retrieved 6,595 articles. Among them, there were 2,105 articles obtained from Chinese (China National Knowledge Infrastructure [CNKI] = 1,563, Wanfang Database = 542) and 4,490 articles written in foreign languages (PubMed = 160, Google Scholar = 4,330). After removing duplicate articles and those that did not meet the required criteria due to unclear distribution information or other reasons, 539 articles were ultimately included in the analysis (S1 Fig). The results revealed the presence of 8 genera and 48 species of ticks distributed in Xinjiang, as shown in Table 1. We selected four dominant tick species for subsequent modeling analysis based on the primary distribution points and collection settings of the tick species. Specifically, we started with *Hyalomma asiaticum* in our investigation, which has been the subject of 77 scholarly articles and can be found in 230 distinct locations. Another species of interest was *Rhipicephalus turanicus*, which has been documented in 35 articles and can be found in 144 different locations. Additionally, we have included *Dermacentor marginatus*, which has been mentioned in 35 articles and can be found in 92 locations. Finally, *Haemaphysalis punctatus* has also been selected for our study, with 21 published articles and a presence in 60 locations. After conducting a detailed analysis of the data, we established a buffer zone of a radius of 10 km around each distribution point with only one point retained using ArcGIS to prevent overfitting. Finally, 120 distribution points of *H. asiaticum*, 86 distribution points of *R. turanicus*, 63 distribution points of *D. marginatus*, and 45 distribution points of *H. punctatus* were determined and included in the prediction model (Fig 1).

### Key environmental variables affecting tick distribution

To improve the accuracy of the MaxEnt model, we selected 19 bioclimatic variables and combined them with altitude, slope, aspect, NDVI, LC, and PD as covariables for model construction (S1 Table). We used jackknife analysis to examine the impact of each variable on the potential suitability areas of the dominant tick species, and the top 4 environmental variables are the Precipitation of Wettest Month (Bio13), Precipitation Seasonality (Bio15), Population Density (PD) and Isothermality (Bio3), which explained a significant quotient of all variables (Fig 2 and S2 Table). The four species of ticks are mainly influenced by Precipitation of Wettest Month (Bio13) and Population Density (PD), and the contribution rates are all far more than 15%. Furthermore, it was observed that, except for *R. turanicus*, Precipitation Seasonality

**Table 1. Distribution records of tick species in Xinjiang by references.**

| Family | Genus | Species | References |
|---|---|---|---|
| Ixodidae | *Dermacentor* | *D. marginatus* | [7,8,18–50] |
| | | *D. nuttalli* | [6–8,20,21,25–28,31–37,39,43,46,47,50–76] |
| | | *D. silvarum* | [8,20,23,27,32–35,37,38,40,46,47,50,52,55,56,59,61,66,70,73,75,77–84] |
| | | *D. niveus* | [8,20,27,30,32,34,37,39,47,50,51,59,70,71,77,81,83,85–97] |
| | | *D. pavlovskyi* | [20,32,55,59,98] |
| | | *D. abaensis* | [20] |
| | | *D. montanus* | [59,65] |
| | | *D. sinicus* | [81] |
| | | *D. reticulatus* | [39] |
| | *Hyalomma* | *H. asiaticum* | [6,7,21,24,26,27,29,30,31,35,36,40–42,45,49,51,54,55,65,71,72,77,91–93,99–129] |
| | | *H. anatolicum* | [20,36,43,47,55,65,71,99,105,109,113,116,119,124,125,127,130–135] |
| | | *H. rufipes* | [20,55,71,115,127] |
| | | *H. dromedarii* | [40,124,125,134] |
| | | *H. asiaticum kozlovi* | [20,27,48,61,70,85,114,117] |
| | | *H. scupense* | [6,8,20,26,27,31,34,37,39–41,50,61,71,72,81,91,93,96,101,105,106,108,115,124,125,130,132,136–139] |
| | | *H. asiaticum caucasicum* | [8] |
| | *Ixodes* | *I. persulcatus* | [26,34,37,39,61,77,79,140–147] |
| | | *I. kaiseri* | [35,44,50,75,148] |
| | | *I. redikorzevi* | [91,93,96] |
| | | *I. redirorzeui* | [96] |
| | | *I. canisuga* | [8,31,44] |
| | | *I. vespertilionis* | [50] |
| | | *I. pavlovskyi* | [8,71] |
| | | *I. arboricola* | [149] |
| | | *I. crenulatus* | [149] |
| | *Haemaphysalis* | *H. punctata* | [8,20,21,25–28,32,36,39,41–43,47,49,50,61,68,71,92,150] |
| | | *H. sulcata* | [20,39,47,49,65,129] |
| | | *H. erinacei* | [20,24,25,30,44,45,91,93,96] |
| | | *H. concinna* | [6,37,41,50,54,61] |
| | | *H. danieli* | [20,22,65,66] |
| | | *H. campanulata* | [34,37] |
| | | *H. warburtoni* | [147] |
| | *Rhipicephalus* | *R. turanicus* | [7,8,20,24–29,31,32,50,65,77,85,89,92,99,110,114,115,127,130,151–160] |
| | | *R. sanguineus* | [8,24,31,36,39,45,50,55,61,71,77,91,96,99,127,153,156,161–166] |
| | | *R. pumilio* | [8,20,39,43,47,65,91,93,96] |
| | | *R. rossicus* | [39,163] |
| | | *R. bursa* | [24,30,31] |
| | | *R. schulzei* | [20] |
| | | *R. haemaphysaloides* | [61] |
| | | *R. microplus* | [31,61,133,167] |
| | *Anomalohimalaya* | *A. lotozkyi* | [65] |
| | | *A. cricetuli* | [149] |
| Argasidae | *Argas* | *A. persicus* | [39,61,65,168,169] |
| | | *A. reflexus* | [170] |
| | | *A. vespertilionis* | [39,171] |
| | *Ornithodoros* | *O. lahorensis* | [49,61,65,77,172–177] |
| | | *O. tartakovskyi* | [61,65,77] |
| | | *O. papillipes* | [93] |

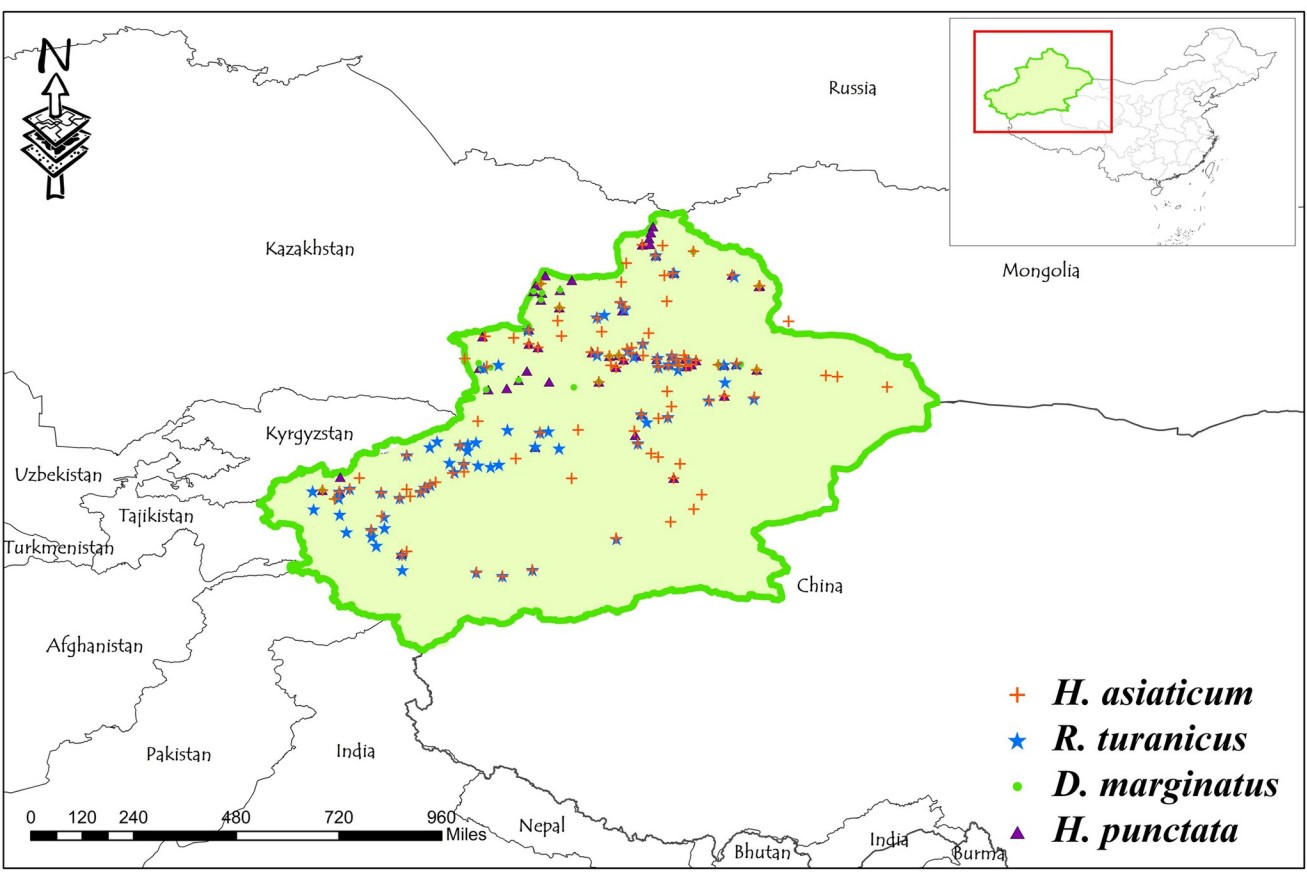

**Fig 1. Distribution map of four dominant tick species in Xinjiang.** The basic map data were obtained from the Department of Natural Resources Standard Map Service System (https://www.webmap.cn/).

(Bio15) exerted a considerable influence on the distribution of the other tick species. This finding highlights the importance of understanding the environmental factors that contribute to the presence and spread of ticks in certain areas.

## Changes in the distribution of suitable habitats of dominant tick species

The distribution of four dominant tick species in the Xinjiang region was determined using the MaxEnt. The suitable habitats for these ticks were divided into four categories. The predictive results indicated that the four tick species models have performed remarkably well with an average AUC value of 0.960, 0.964, 0.956, and 0.986, which suggested that the predictive accuracy of the models was excellent. The suitable habitat for *H. asiaticum* was mainly found in areas such as Tacheng, Altay, and Kizilsu Kirghiz Autonomous Prefecture, while unsuitable habitat was primarily distributed in regions like Hami and Bayingolin Mongol Autonomous Prefecture. In the future climate scenario, the potential suitability areas for *H. asiaticum* would fluctuate, with an overall increase in proportion compared to the current model (Fig 3). By 2081–2100, the potential suitability areas will increase to 297710.11 km$^2$, expanding further into areas such as Hami and Hetian. However, there will be a loss of potential suitability areas in certain parts of Kashgar (Fig 4A).

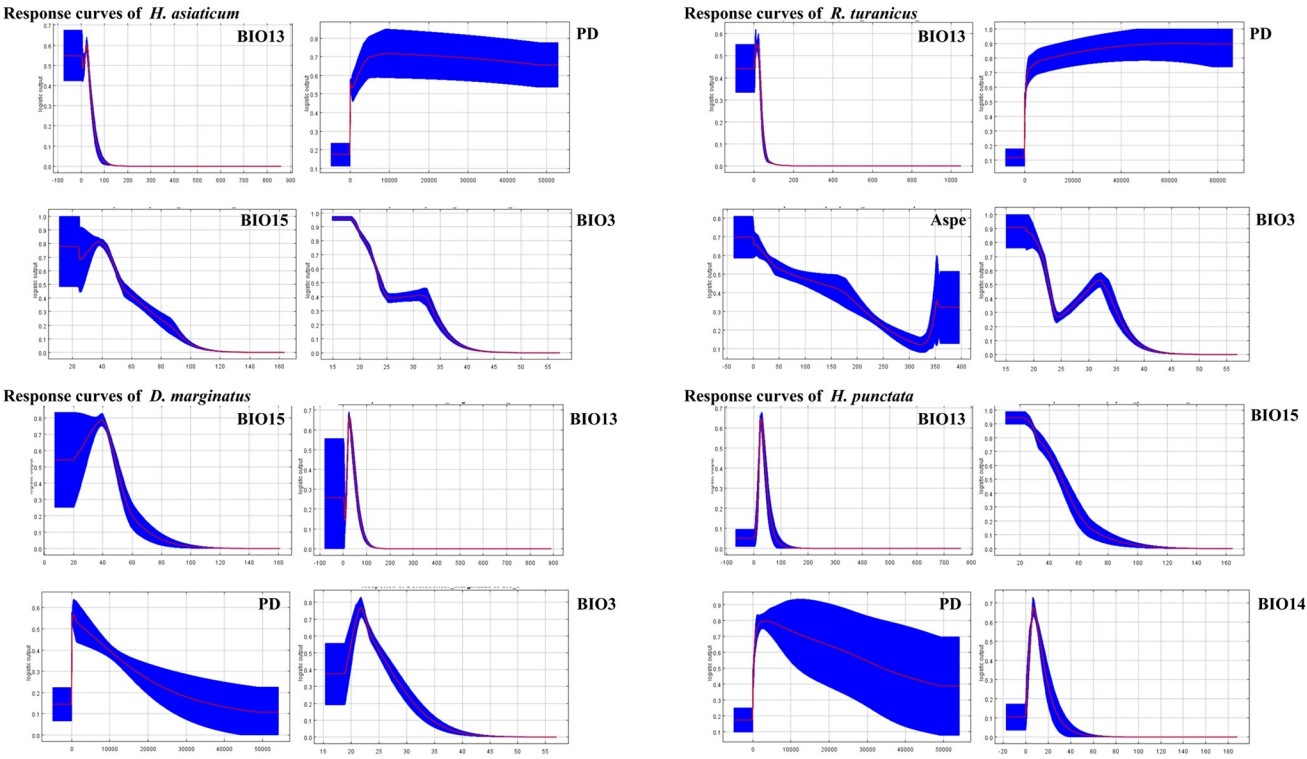

**Fig 2. Response curves of the Maxent model representing *H. asiaticum*, *R. turanicus*, *D. marginatus* and *H. punctata* probability of occurrence against the top four environmental predictors.**

The potential suitability areas for *R. turanicus* were mainly distributed in the Tacheng region, Changji Hui Autonomous Prefecture, and Kashgar region. In contrast, Hami region, Bayingolin Mongol Autonomous Prefecture, and Hotan region were the unsuitability areas (Fig 5). Under future climate scenarios, the potential suitability areas for *R. turanicus* increased to 83921.92 km$^2$ by 2081–2100, with new potential suitability areas distributed in Altay, Tacheng, and Changji Hui Autonomous Prefecture (Fig 4B).

The potential suitability areas for *D. marginatus* were mainly distributed in Tacheng, Kashgar, and Kizilsu Kirghiz Autonomous Prefectures, while the unsuitability areas are distributed primarily in the Hami and Bayingolin Mongol Autonomous Prefecture (Fig 6). Under future climate scenarios, the potential suitability areas were expected to experience an expansion of 86,087.65 km$^2$ by 2081–2100 and mainly concentrated in certain areas such as Tacheng and Changji Hui Autonomous Prefecture. The diminished suitability areas comprised 50% of the increased suitability areas, predominantly concentrated in some parts of Altay and Kizilsu Kirkirz Autonomous Prefecture (Fig 4C).

The distribution of potential suitability areas for *H. punctata* was primarily concentrated in Tacheng, Kashgar, and Kizilsu Kirghiz Autonomous Prefectures, while the unsuitability areas were mainly found in Hami and Bayingolin Mongol Autonomous Prefecture (Fig 7). It is projected that the potential habitats (both high- and medium-suitability area) will decrease by 71778.38 km$^2$ by 2081–2100 under future climate scenarios, primarily concentrated in Kizilsu Kirghiz Autonomous Prefecture, but with an expansion in Aksu and Tacheng regions (Fig 4D).

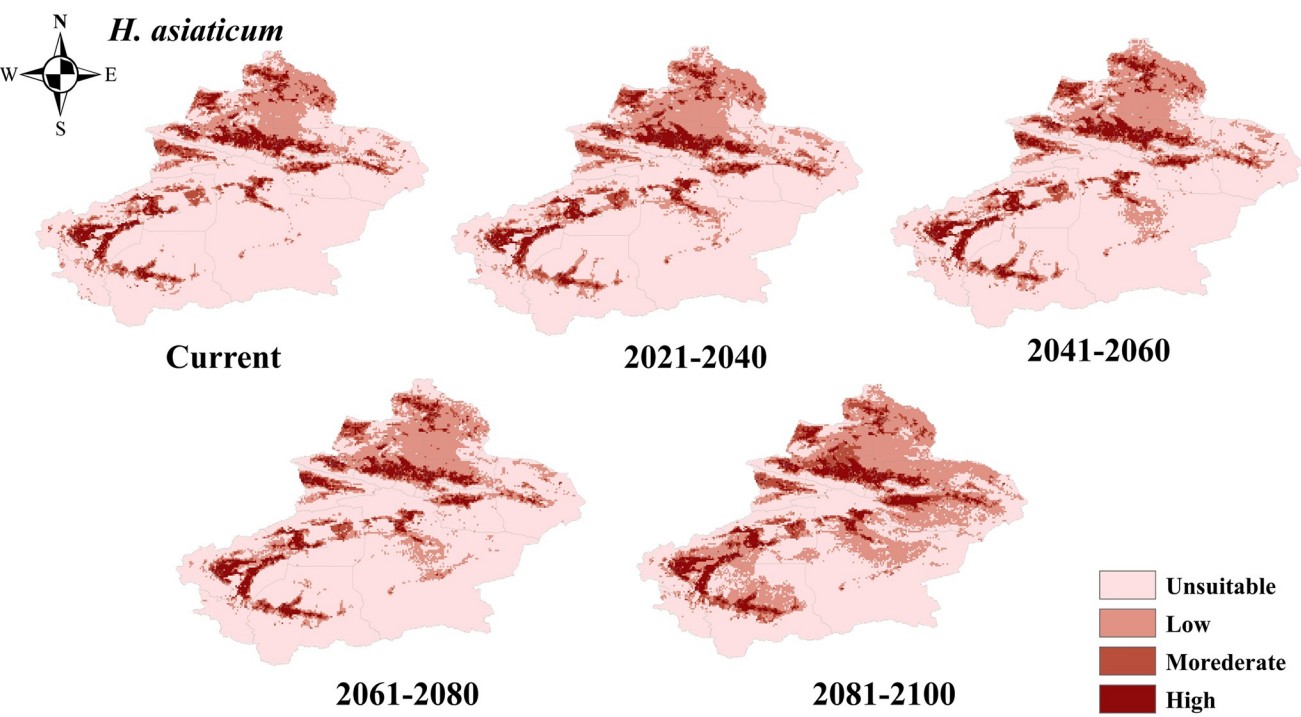

**Fig 3. Projected changes in the potential distribution of *H. asiaticum* in Xinjiang at different periods.** The basic map data were obtained from the Department of Natural Resources Standard Map Service System (https://www.webmap.cn/).

### Centroid shift trend of dominant tick species in future climatic conditions

Fig 8 shows the potential shifts in the centroid of four tick species habitats from 2081–2100 compared to their current condition (S3 Table). The centroid of *H. asiaticum* has been observed to shift east-northward by 103.43 km within the Tacheng region. The longitude of the centroid remains relatively stable, while the latitudes exhibited slight variations in spacing (from 43°38′24″N to 42°39′15″N). The centroid of *R. turanicus* moved 80.36 km from the Aksu region to Bayingolin Mongol Autonomous Prefecture, with a latitude and longitude change of about 35' (from 42°10′12″N to 42°43′12″N). The centroid of *D. marginatus* and *H. punctata* showed a southeastward movement trend, with little change in latitude and longitude. Specifically, the centroid of *D. marginatus* moved 56.93 km from the Tacheng region to Bayingolin Mongol Autonomous Prefecture, while the centroid of *H. punctata* slightly moved 18.02 km within the Tacheng region.

## Discussion

Ticks are distributed globally, and with the expansion of human activity ranges and changes in ecological environments, the opportunities for contact between ticks and humans and animals increase, leading to an increased risk of infection with tick-borne diseases. Moreover, it is important to note that ticks can also be transmitted through the movement of infected animals, such as livestock and wildlife, leading to spreading tick-borne diseases across borders and even globally [178]. Climate change has exacerbated the spread of ticks worldwide, with theories suggesting higher tick proliferation rates, extended transmission seasons, and climate-related migrations. Therefore, this study utilizes the latest version of climate factors and

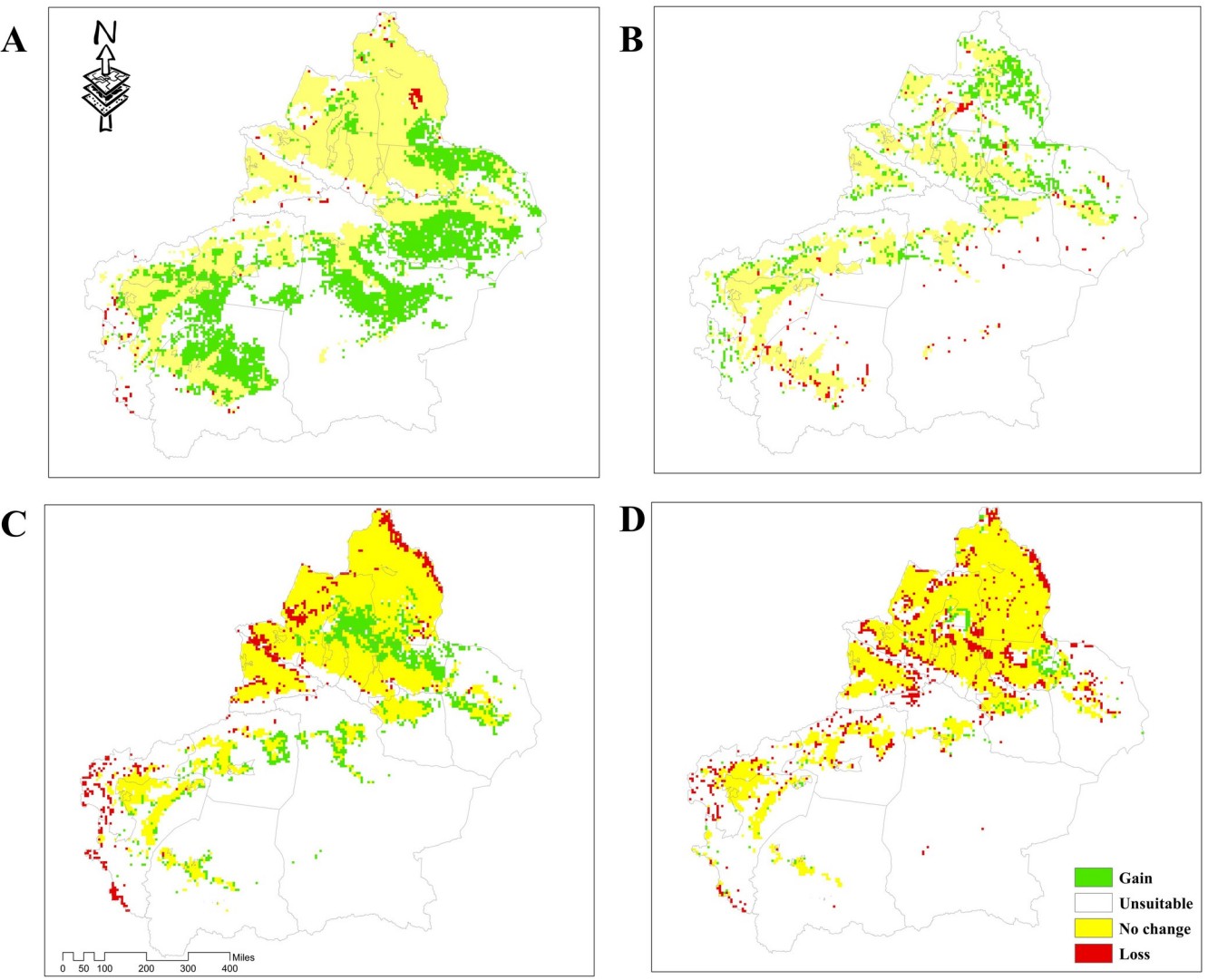

**Fig 4. Changes in the potential suitability areas for the four dominant tick species under current and 2081–2100.** The basic map data were obtained from the Department of Natural Resources Standard Map Service System (https://www.webmap.cn/).

distribution data to provide an updated spatial layout of the species' distribution and the potential determining environmental factors under different scenarios. After conducting a literature review, we identified and depicted eight genera and 48 species of ticks reported in Xinjiang over the past 60 years. We also predicted suitability areas using the MaxEnt model and analyzed the influential environmental variables under different scenarios. The findings from this study will be crucial in forecasting the future distribution patterns of the species, as well as in monitoring and assessing the risk of disease transmission across borders in the region.

The distribution of the dominant tick species in the Xinjiang region is mainly influenced by the variables of Precipitation of Wettest Month (Bio13), Precipitation Seasonality (Bio15), and Population Density (PD), with their contribution rates exceeding 17%. When the Precipitation of Wettest Month (Bio13) values are 23.32, 22.04, 24.98, and 26.77 mm, respectively, the distribution probabilities of the four tick species reach their maximum, which may be a key reason for the overlapping distribution zones of these four tick species. In 2020, Yang et al. conducted

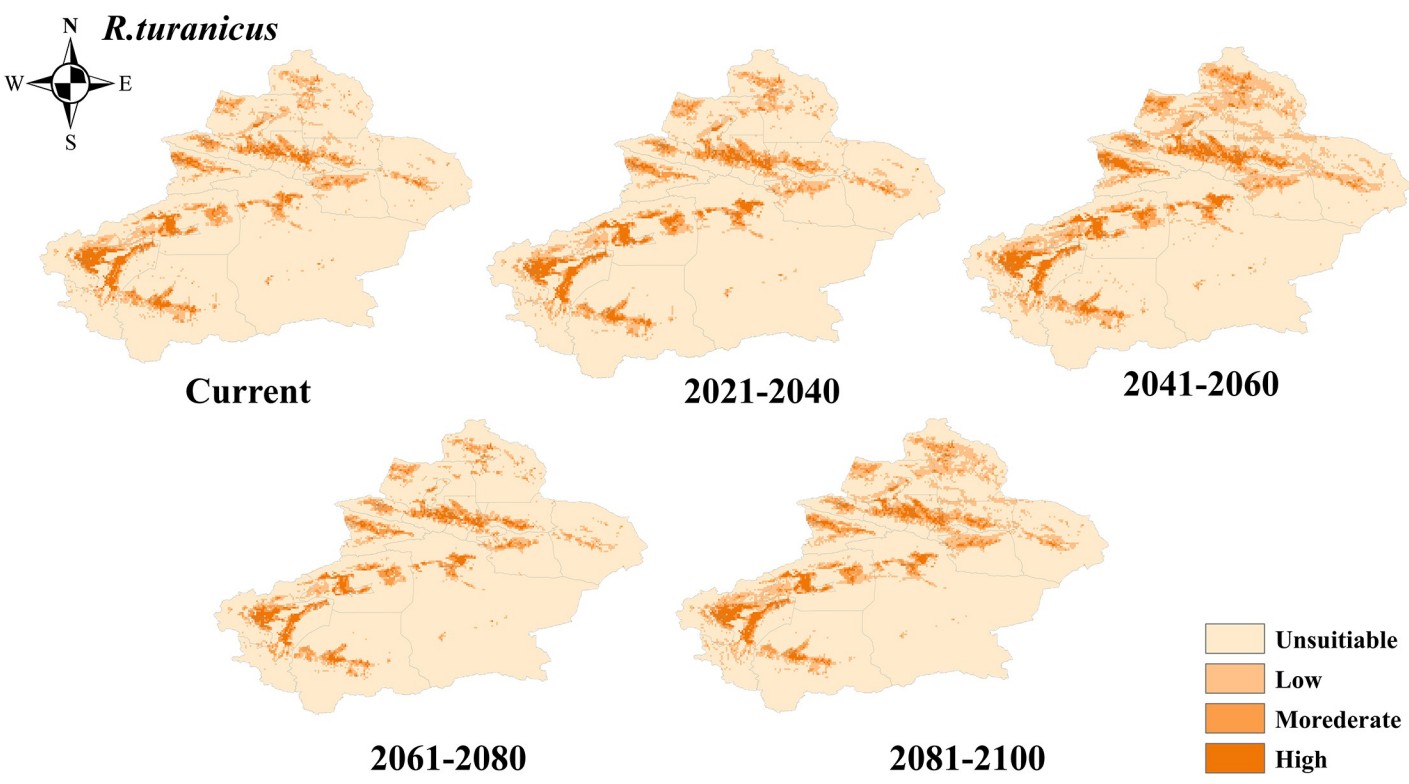

**Fig 5. Projected changes in the potential distribution of *R. turanicus* in Xinjiang at different periods.** The basic map data were obtained from the Department of Natural Resources Standard Map Service System (https://www.webmap.cn/).

a nationwide survey of *D. marginatus* in China and found that Bio13 significantly impacted their distribution. Meanwhile, this study pointed out that *D. marginatus* is more sensitive to precipitation than other tick species studied [179]. In addition, Precipitation Seasonality (Bio15) also has a significant impact on the distribution of *H. asiaticum*, *D. marginatus*, and *H. punctata*, which is similar to the results of the global model conducted by Wang et al. in 2019 and our previous predictive model for dominant ticks in Inner Mongolia [180,181]. In this study, the distribution of *R. turanicus* is not affected by Bio15, which may be related to its primary distribution in desert or semi-desert areas. In addition, the distribution probabilities of four tick species increased with Population Density (PD), with the most notable being the *R. turanicus*, which reached a maximum distribution probability of 0.90 when PD was 65110.53 people/km$^2$. According to a study on the potential distribution areas of *Haemaphysalis longicornis* in China, the PD contribution rate was found to be 23.1%, indicating that PD is an important factor affecting the species' distribution [182]. The alteration in the type and magnitude of human activities can considerably influence the distribution of ticks and the transmission of tick-borne diseases. Comprehending these associations is of utmost importance in devising efficacious approaches to diminish the hazard of tick-borne ailments in both humans and animals.

The distribution model presented in this study demonstrates that the primary suitable habitats for the four tick species are currently and will continue to be predominantly concentrated in the northern region of Xinjiang. Our research findings align with the predictive outcomes reported by Huercha et al. (2020) regarding the suitable habitat for marginal ticks in Xinjiang [183]. The northern part of Xinjiang is characterized by diverse grassland resources, which

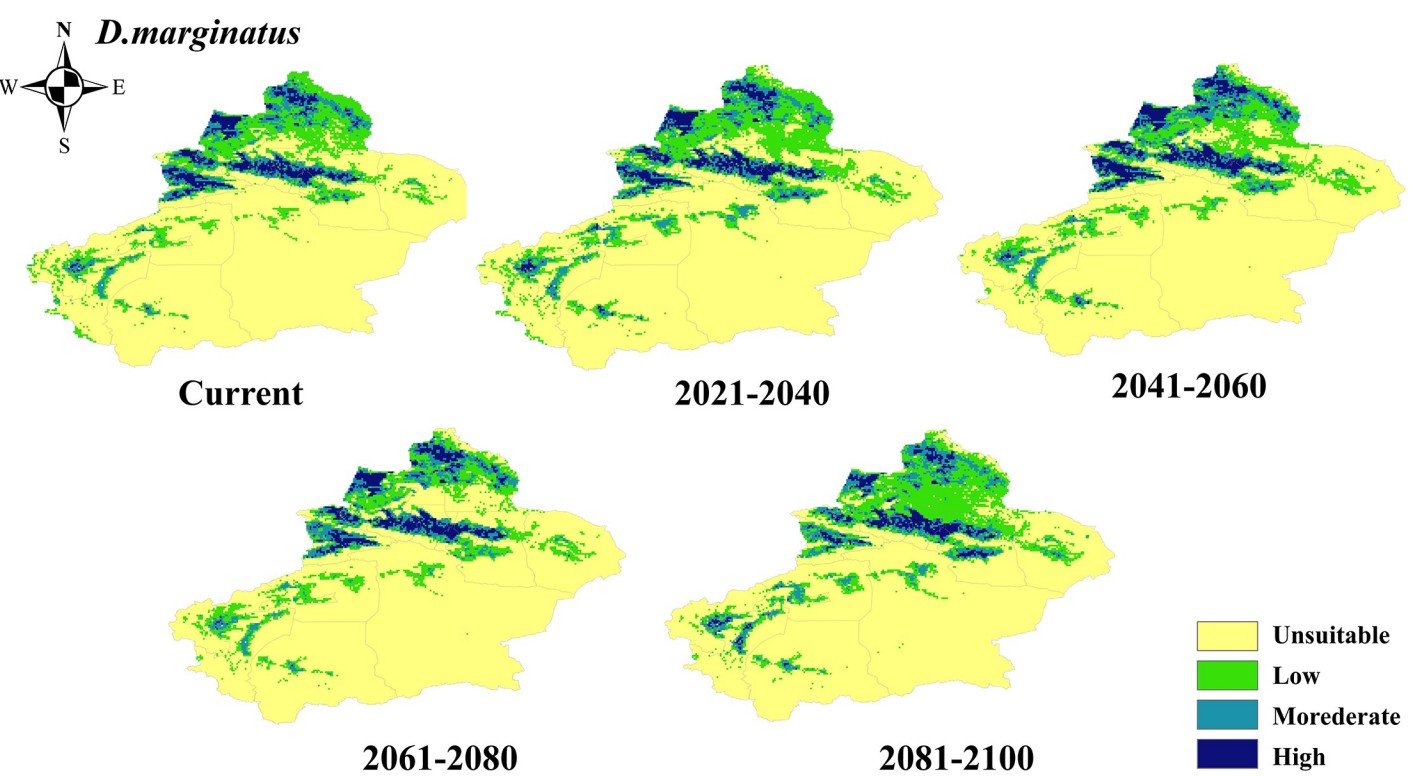

**Fig 6. Projected changes in the potential distribution of *D. marginatus* in Xinjiang at different periods.** The basic map data were obtained from the Department of Natural Resources Standard Map Service System (https://www.webmap.cn/).

serve as the primary habitat for free-living ticks. Additionally, it is an important location for nomadic activities. Ticks that parasitize animals on their surface can also migrate to new suitable habitats alongside animal migration, thereby facilitating the spread of ticks and tick-borne diseases. In addition to the factors mentioned above, the region has multiple wild rodent species, further increasing the potential hosts and dispersal risks of ticks. This study analyzed prediction models for the suitability areas of four dominant tick species across four future periods. The findings revealed that all four species exhibited an expansion of their suitability areas towards the northern direction, albeit to varying degrees. This observation aligns with the conclusions drawn by Yang et al. and He et al. [184,185]. Furthermore, it was observed that the suitability areas for the dominant tick species increased from 2081 to 2100, except for *H. punctata*, which experienced a decrease primarily in the southern part of Xinjiang. Additionally, there was a notable shift towards the northeastern direction in the centroid of habitats for *H. asiaticum* and *R. turanicus*.

Given the increasing threat of tick-borne diseases, exacerbated by global climate change and expanding human activities, understanding the distribution of dominant tick species under current and future climatic conditions becomes paramount. The MaxEnt model has emerged as a powerful tool with high prediction accuracy in various ecological modeling, particularly in predicting species distributions and identifying potential suitability areas [186]. This study delves into the application of the MaxEnt model in predicting the distribution of tick species in Xinjiang and its implications for global health. The model effectively integrates vital climatic and environmental factors to provide a comprehensive understanding of tick distribution patterns under various future scenarios. While MaxEnt is flexible in terms of data

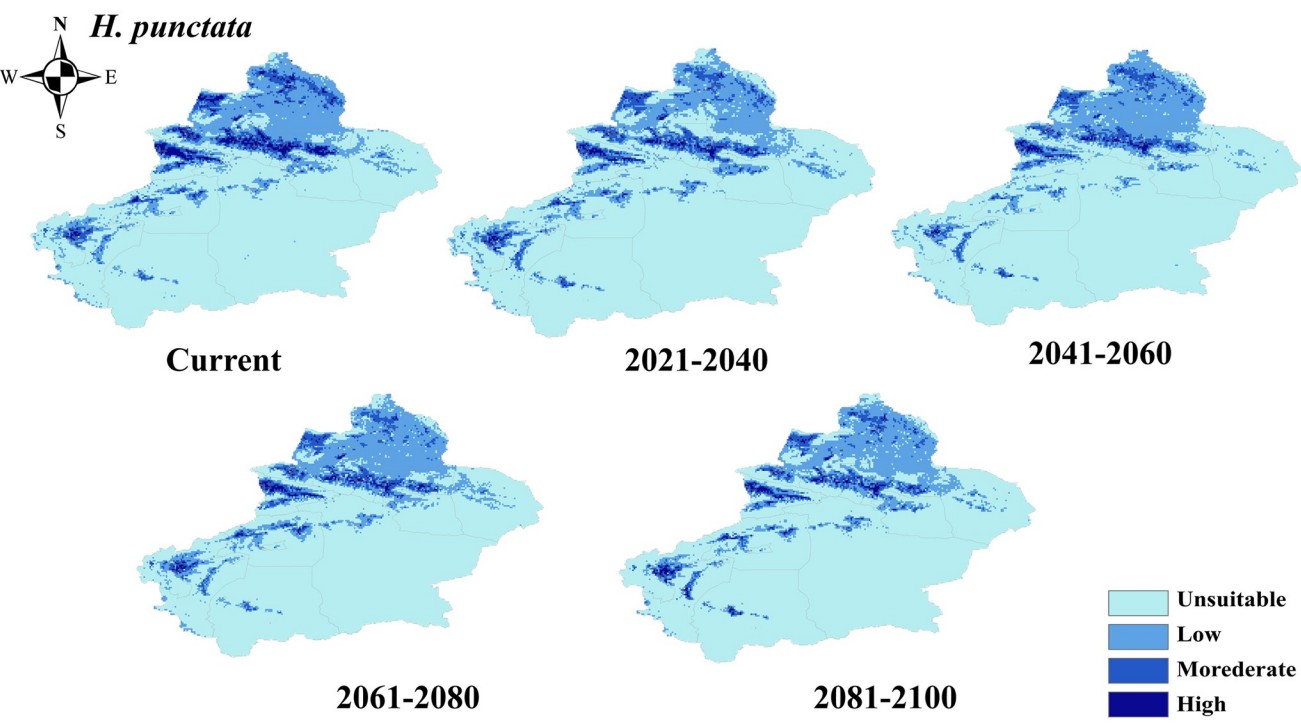

**Fig 7. Projected changes in the potential distribution of *H. punctata* in Xinjiang at different periods.** The basic map data were obtained from the Department of Natural Resources Standard Map Service System (https://www.webmap.cn/).

requirements, the accuracy of its predictions is heavily reliant on the quality and comprehensiveness of the input data.

Moreover, it's crucial to be careful that MaxEnt is susceptible to overfitting, especially when too many environmental variables are included, or the model is too complex for the available data [187,188]. Therefore, this study still has some limitations. Firstly, the selection of distribution points is mainly based on published literature data. Although we have eliminated duplicate samples and corrected the geographical information bias in the tick distribution dataset, this is only based on publicly available tick distribution information within the Xinjiang region. Tick distribution may also exist in areas that have not been reported in the literature. In the predictive model, only the potential suitability areas for the given species are provided, which may overlook the actual distribution of some ticks and fail to reflect the population density changes of ticks. In addition to the 19 conventional climate factors, we have also included major landform variables such as slope, aspect, and altitude, as well as population density, but there are still some variables that have not been included that affect the distribution of mosquitoes, such as animal hosts, soil pH, and humidity. Moreover, we can also consider extracting the six types of land cover separately to analyze their individual impacts in the future.

## Conclusion

In conclusion, this study depicted the preliminary distribution map of tick species in Xinjiang over 60 years and construct potential spatial distribution models for current and future dominant tick species. Analyzing critical environmental factors and predicting future trends in suitable habitats provides valuable insights into tick distribution and risk assessment on cross-

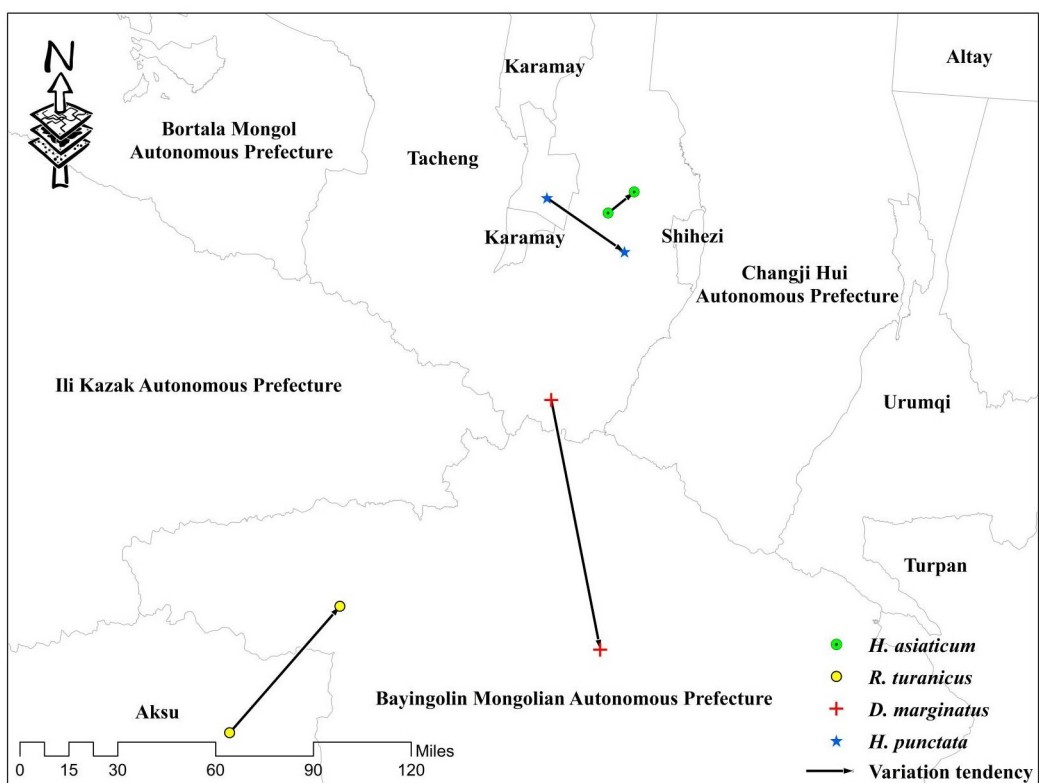

**Fig 8. Changes in the centroid of the potential suitability areas for dominant tick species.** The basic map data were obtained from the Department of Natural Resources Standard Map Service System (https://www.webmap.cn/).

border vector disease transmission. The findings have significant implications for disease prevention and control strategies in Xinjiang as well as countries along the BRI.

## Supporting information

**S1 Table. Environmental and climatic variables for the four dominant tick species distribution models by Maxent.**
(XLSX)

**S2 Table. Key variables inferred from the MaxEnt model influencing ticks' distribution.**
(XLSX)

**S3 Table. Centroid shift trend of dominant tick species under current and 2081–2100.**
(XLSX)

**S1 Fig. Flow diagram of literature search and inclusion.**
(TIF)

## Author Contributions

**Conceptualization:** Jian Li, Wei Hu, Xinyu Feng.

**Data curation:** Rui Ma, Chunfu Li, Ai Gao, Na Jiang.

**Formal analysis:** Rui Ma, Chunfu Li.

**Funding acquisition:** Wei Hu.

**Investigation:** Rui Ma, Ai Gao, Na Jiang.

**Methodology:** Rui Ma, Chunfu Li, Ai Gao, Na Jiang.

**Project administration:** Jian Li, Wei Hu, Xinyu Feng.

**Resources:** Jian Li, Wei Hu, Xinyu Feng.

**Software:** Rui Ma, Chunfu Li, Ai Gao, Na Jiang.

**Supervision:** Jian Li, Wei Hu, Xinyu Feng.

**Validation:** Rui Ma, Ai Gao, Na Jiang.

**Visualization:** Rui Ma, Chunfu Li, Ai Gao, Na Jiang.

**Writing – original draft:** Rui Ma, Xinyu Feng.

**Writing – review & editing:** Chunfu Li, Ai Gao, Na Jiang, Jian Li, Wei Hu, Xinyu Feng.

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
