## [Decision Letter · Decision Letter 0]

15 Feb 2024

Dear Dr. Feng,

Thank you very much for submitting your manuscript "Tick species diversity and potential distribution alternation of dominant ticks under different climate scenarios in Xinjiang, China" for consideration at PLOS Neglected Tropical Diseases. As with all papers reviewed by the journal, your manuscript was reviewed by members of the editorial board and by independent reviewers. In light of the reviews (below this email), we would like to invite the resubmission of a significantly-revised version that takes into account the reviewers' comments. 

In particular, please address the significant points and concerns raised by Reviewer 1. The additional comments from Reviewer 2 will also need to be addressed. Please ensure that you address each point clearly and thoroughly. Additional discussion of the potential applicability of the strengths and weaknesses of your approaches on the global rather than regional scale would increase audience interest.

We cannot make any decision about publication until we have seen the revised manuscript and your response to the reviewers' comments. Your revised manuscript is also likely to be sent to reviewers for further evaluation.

Sincerely,

Jenifer Coburn, PhD

Academic Editor

Paul Mireji

Section Editor

Reviewer's Responses to Questions

**Key Review Criteria Required for Acceptance?**

**Methods**

-Are the objectives of the study clearly articulated with a clear testable hypothesis stated?

-Is the study design appropriate to address the stated objectives?

-Is the population clearly described and appropriate for the hypothesis being tested?

-Is the sample size sufficient to ensure adequate power to address the hypothesis being tested?

-Were correct statistical analysis used to support conclusions?

-Are there concerns about ethical or regulatory requirements being met?

Reviewer #1: Readers likely do not know what the “China National Knowledge Infrastructure” provides – so this needs to be explained.

In some cases “the coordinates of the distribution points were determined using the coordinate picking function in Google Maps based on the geographical location” but this needs more explanation – there must be some criterion on which to base use of such articles – maybe if they mentioned a particular town or municipality?

“Only one distribution point was retained in each 10 km × 10 km grid” – what does this mean as in the previous sentence it was mentioned that a buffer of 10km was used?

A correlation coefficient of 0.9 is a very low bar for eliminating possibly correlated explanatory variables. Often 0.6 is used.

The WorldClim database is used for climate data for the MaxEnt model and spatial resolution of this is provided. However WorldClim is an interpolation of climate station data – so the article needs to identify how many climate stations in the study region, that contributed to WorldClim are present in the study region. Confidence in the climate data used would be very different if there were 3 or 300 meteorological stations providing data.

A table identifying the geographic resolution of explanatory variable data and tick data would dhelp the reader understand the relationships amongst these data.

Reviewer #2: Please provide the languages and search terms used for the literature search described in lines 84-87.

Line 114, please change "the main factor" to "an important factor". Climate is not the single main factor influencing tick distribution.

**Results**

-Does the analysis presented match the analysis plan?

-Are the results clearly and completely presented?

-Are the figures (Tables, Images) of sufficient quality for clarity?

Reviewer #1: It is stated that there was a ‘top 4’ variables used – but only 3 are mentioned.

It seems that the only climatic determinant of tick distributions is precipitation, which is much more uncertain in climate model outputs than temperature.

Reviewer #2: Results are appropriate

**Conclusions**

-Are the conclusions supported by the data presented?

-Are the limitations of analysis clearly described?

-Do the authors discuss how these data can be helpful to advance our understanding of the topic under study?

-Is public health relevance addressed?

Reviewer #1: Throughout it needs to be clear that collecting information from literature only provides possible distributions (the article does not provide an ‘up-to-date map of tick distribution in Xinjiang’ as mentioned in the abstract) – not actual distributions, and that extensive field study is needed to confirm or deny findings in the article.

Reviewer #2: Discussion and conclusion are appropriate

**Editorial and Data Presentation Modifications?**

Reviewer #1: (No Response)

Reviewer #2: The figures are a little blurry and authors will need to upload high-resolution map images. 

Overall, the English writing and grammar are good; however, small editorial changes will need to be made. For example, several terms are capitalized when they should be lowercase.

**Summary and General Comments**

Reviewer #1: Major comments:

The article obtains from the literature possible tick distributions, and from these, possible future distributions are projected. Throughout it needs to be clear that collecting information from literature only provides possible distributions (the article does not provide an ‘up-to-date map of tick distribution in Xinjiang’ as mentioned in the abstract) – not actual distributions, and that extensive field study is needed to confirm or deny findings in the article.

The treatment of projected future climate is very limited. There is only one greenhouse gas concentrations scenario (when normally a range would be included), and no accounting for variations amongst models in the CMIP6 ensemble. These sources of variations should be included.

How results from MaxEnt models are used to provide climate envelopes for tick species, that are then used to project future tick distributions, is poorly explained in the methods.

MaxEnt is only one amongst a number of methods that are used to estimate determinants of the ecological niche of species – explanation of the selection of this method, and rejection of others, is needed.

Minor comments

The article needs a check on English language – overall it is very good but the abstract needs particular attention.

Abstract:

Ticks are not reservoirs.

Data, being plural, ‘were’ rather than ‘was’.

Some specifics are needed on key tick species rather than simply saying they ‘fluctuated’.

Introduction:

Please include the tick species involved in TBEV-2871 transmission.

It is true that “Previous studies show that the distribution of ticks is closely related to the natural environment and exhibits distinct regional and seasonal characteristics” but the literature cited is very limited.

The objective “to depict the distribution map of tick species in Xinjiang over a period of 60 years” is not correct – the objective is to identify possible distributions from 60 years of literature.

Reviewer #2: The authors describe a new multi-country highway system that has the potential to inadvertently introduce new tick species into the region of Xinjiang, China. This article is a new contribution to the scientific literature and has broader implications for other countries who are planning large-scale transportation infrastructure projects. The authors leveraged the historical literature to inform climate change models to predict future expansion of vector-specific habitable areas in the region.

PLOS authors have the option to publish the peer review history of their article (what does this mean?). If published, this will include your full peer review and any attached files.

Reviewer #1: No

Reviewer #2: No
---

## [Editor Report · Decision Letter 1]

27 Mar 2024

Dear Dr. Feng,

We are pleased to inform you that your manuscript 'Tick species diversity and potential distribution alternation of dominant ticks under different climate scenarios in Xinjiang, China' has been provisionally accepted for publication in PLOS Neglected Tropical Diseases.

Best regards,

Jenifer Coburn, PhD

Academic Editor

Paul Mireji

Section Editor

---

## [Editor Report · Acceptance letter]

12 Apr 2024

Dear Dr. Feng,

We are delighted to inform you that your manuscript, "Tick species diversity and potential distribution alternation of dominant ticks under different climate scenarios in Xinjiang, China," has been formally accepted for publication in PLOS Neglected Tropical Diseases.

Best regards,

Shaden Kamhawi

co-Editor-in-Chief

Paul Brindley

co-Editor-in-Chief
